# A Screening Test for Early Diagnosis of Microcellular Bronchopulmonary Cancer-Pilot Study

**DOI:** 10.3390/jcm9010076

**Published:** 2019-12-27

**Authors:** Claudiu-Eduard Nistor, Raluca-Ioana Stefan-van Staden, Adrian Vasile Dumitru, Camelia Stanciu Găvan

**Affiliations:** 1Department of Thoracic Surgery the Central Military Emergency University Hospital Bucharest, University of Medicine and Pharmacy “Carol Davila”, Bucharest 010825, Romania; 2Laboratory of Electrochemistry and PATLAB Bucharest, National Institute of Research for Electrochemistry and Condensed Matter, Bucharest 060021, Romania; 3Department of Pathological Anatomy Emergency University Hospital Bucharest, University of Medicine and Pharmacy “Carol Davila”, Bucharest 050098, Romania; dr.adidumitru@yahoo.com

**Keywords:** Microcellular Bronchopulmonary Cancer, NSE, CEA, IHC, stochastic sensors, ELISA

## Abstract

Introduction: According to WHO, in worldwide cancer mortality statistics, the first place is occupied by bronchopulmonary cancer. This reason has led us to carry out the present pilot study, was with the participation of the Clinics of Carol Davila University of Medicine and Pharmacy Bucharest in order to apply a technique developed earlier by Stefan-van Staden, for early detection of this type of cancer, initiate a personalized diagnosis, and implicitly apply a personalized treatment in order to increase the life expectancy among these patients. In recent years, there has been a tendency to find fast non-invasive screening methods for the early diagnosis of cancer. Therefore, the present pilot study proposed simultaneous detection of tumor markers (NSE and CEA) by different methods: (1) ELISA kits, (2) the method developed earlier by Stefan-van Staden—which used stochastic sensors, and (3) IHC. All selected patients selected by Dr Claudiu-Eduard Nistor, were suspected of microcellular bronchopulmonary cancer. Tumor tissue samples were collected by conventional and minimally invasive surgical techniques. The results obtained for the detection of markers in blood using ELISA, and stochastic methods (based on stochastic sensors) were correlated with the results obtained using anatomopathological and immunohistochemical analysis of the tumor tissues. Experimental: Stochastic sensors have been used to analyze NSE in blood samples and whole tissues. The IHC was performed for analyzing tumor tissue using standard procedures. ELISA has been used as a standard method to determine specific biomarkers in whole blood samples. Results and discussion: A good correlation was found for results obtained using stochastic and ELISA methods, and IHC for blood and tissue analysis. Statistical evaluation of the data showed that the results of whole blood analysis are correlating very good with the analysis of pulmonary tumor tissue. Therefore, the stochastic method can be used for the detection and for the pursuit of therapeutic efficiency. Conclusions: The data obtained, as well as the statistics, showed that the proposed method can be used as a screening method for fast and early detection of microcellular bronchopulmonary, being minim invasive. It can also be used for monitoring the therapeutic efficiency of the prescribed medication.

## 1. Introduction

Lung cancer is a matter of utmost importance in oncological pathology worldwide. This disease is one of the main causes of death in both men and women worldwide, according to WHO [1]. Most bronchopulmonary cancer cases were detected in advanced stages of the disease with very low chances of survival. Early detection of these cancers is the most promising approach for increasing the survival of patients diagnosed with lung cancer because the initiation of early treatment leads to increased survival percentages.

Neuron-specific enolase (NSE) is a tumor biomarker found in patients with microcellular lung cancer [2]. Up to 70% of patients with bronchopulmonary cancer have elevated serum NSE concentrations at diagnosis, and approximately 90% of patients with advanced bronchopulmonary cancer will have serum levels above the healthy reference range. Other neuroendocrine tumors with frequent expression of NSE include carcinoids (up to 66% of cases), islet cell tumors (typically <40% of cases), and neuroblastoma (exact frequency of NSE expression unknown). NSE levels in NSE-secreting neoplasms correlate with tumor mass and tumor metabolic activity. High levels have therefore some negative prognostic value. Falling or rising levels are often correlated with tumor shrinkage or recurrence, respectively. There was a good correlation between the NSE levels, the clinical stage and the degree of enlargement of the disease. Therefore, this biomarker can be considered for a screening test that may facilitate the early detection of microcellular lung cancer. High concentrations of carcinoembryonic antigen (CEA) (>20 ng/mL) in a patient with compatible symptoms are suggestive for the presence of cancer and suggest metastases [3]. NSE is a sensitive, specific and reliable diagnostic biomarker for small cell lung cancer, but its presence is also increased in other malignant tumors such as melanoma, neuroblastoma, hormone-resistant prostate cancer, and the Semioma. Values recorded for healthy patients are in the range of 1.5–15 ng/mL. Values representing moderate increases in the serum concentration of this marker are 15–40 ng/mL. Simultaneous detection of NSE and CEA using screening tests shown previously the early detection of microcellular lung cancer [4]. For the present study, the tumor tissue samples were collected by classical and minimal invasive surgical techniques.

The aim of this study was to comparatively assess the levels of NSE and CEA using three different methods of analysis: (1) ELISA kit (the accredited method of analysis), (2) the stochastic method (developed earlier by Stefan-van Staden), and (3) IHC-for tissue samples, in order to be able to implement after clinical studies the stochastic method as a very fast, reliable and cheap method of whole blood screening in order to: (1) facilitate early detection of microcellular lung cancer, and (2) facilitate the evaluation of the efficiency of treatment of patients with microcellular lung cancer.

In terms of clinical implications in the diagnosis and personalized treatment of the microcellular lung cancer, the screening test of whole blood for these tumor biomarkers aims at the early effective personalized treatment and the pursuit of therapeutic efficiency by monitoring the levels of NSE and CEA found in whole blood.

Stochastic sensors were intensively used for biomedical analysis due to their advantages: no treatment is needed for the whole blood or tissue before analysis; the matrix of the samples did not interfere in the measurements—the cost is very low, and they can be used for the assay of more than one biomarker in the same run. The working principle of the stochastic sensors is based on channel conductivity, and therefore a qualitative analysis (based on the finding of the signature of biomarker in the recorded diagram) and quantitative analysis can be performed [4].

The Pilot study was performed using 45 patients, 22 women and 23 men, between the 2015–2018 timeframe. Most patients were detected in stages III and IV of the disease—the tumor was detected using imagistic techniques suspects of Microcellular Bronchopulmonary Cancer; the tumor tissue samples were collected by classical and minimal invasive surgical techniques; no anatomopathological diagnosis was performed initially.

## 2. Experimental Part

The study was carried out using the following biological samples: whole blood, serum, and tissue samples. ELISA technique was used for the determination of NSE and CEA in serum samples. Stochastic sensors were used for the assay of NSE and CEA in whole blood samples and tumor tissue; and assay IHC in tumor tissue—NSE.

The initial selection of lung cancer patients from the analyzed batch was performed based on CT imaging examination. Immunohistochemical analysis of tumor tissue were collected in the thoracic surgery section by Dr. Claudiu-Eduard Nistor, was performed.

### 2.1. Instrumentation

The stochastic method was performed using a Potentiostat/PGSTAT 12 (Metrohm, Utrecht, Netherlands) software version 4.9. IHC and ELISA were performed in accredited clinical laboratories, with accredited instrumentation.

Immunohistochemical Diagnosis-IHC for each diagnostic sample from tumor tissue a representative paraffin block based on the staining of Hematoxyillin-Eozine for Immunohistochemistry is selected. It is stained with a panel consisting of not more than 38 antibodies, of which eventually 27 were taken into consideration for analysis, after removal of missing data. Immunohistochemia is the most accurate branch of the technique of histology that brings certainty that confirms, infirm or supports anatomopathological diagnosis [5].

ELISA technique—(Enzyme Linked Immunosorbet Assay) is currently one of the most commonly used methods, identification and quantitative appreciation for antibodies, antigens, hormones, cytokines and a wide range of other molecules, including synthetic peptides. In Our study, ELISA tests for the CEA and NSE were carried out according to the manufacturer’s instructions with minor modifications. The samples of these markers were worked in the laboratory of the Clinics of Carol Davila University of Medicine and Pharmacy Bucharest as well as in other laboratories outside the hospital, when the kits were insufficient in the hospital’s laboratories. ROCHE analysers and ELISA kits have been used, on 96 Breakable Wells coated. The results of ELISA tests for NSE and CEA that obtained in the group of patients taken into study were positive for 40 cases, and in a number of 5 cases the results were negative [6].

### 2.2. Preparation of Biological Samples

Blood and tissue samples were taken from the Clinics of Carol Davila University of Medicine and Pharmacy Bucharest, following the protocol approved by the Ethics Commission of the University of Medicine and Pharmacy Carol Davila, from Bucharest, having the registration number 75/2015. Informed consent was obtained from all patients selected, diagnosed with lung cancer or suspected of pulmonary cancers. Biological samples were used for screening for NSE and CEA. No treatment was applied to the blood and tissue samples when the stochastic method was used. Serum samples were used for the assay of CEA and NSE using ELISA.

Tissue samples obtained from patients with lung cancer, suspects of Microcellular Bronchopulmonary Cancer—collected by classical and minimal invasive surgical techniques were fixed with 10% buffered formalin and were processed by conventional histopathological methods using paraffin incorporation, sectioning, and staining with hematoxylin-eosin. Immunohistochemical tests have also been performed using the NSE antibody.

The group of patients who formed the basis of the present pilot study comprised on 45 patients, from which 22 were women and 23 men, with ages between 38 and 87 years, with the localization of the tumor as following: 18 patients in the left lung and 27 in the right lung.

### 2.3. Determination of CEA and NSE Using ELISA, IHC, and Stochastic Method

In the present study, ELISA tests for the CEA and NSE were carried out according to the manufacturer’s instructions. The assay of CEA and NSE from the samples was performed in the laboratory of the University of Medicine and Pharmacy “Carol Davila”, Bucharest, Romania, as well as in other accredited laboratories outside the hospital. ROCHE analyzers and ELISA kits have been used on 96 breakable wells [6].

Histopathological analysis was performed by examination with optical microscopy of sections obtained from paraffin blocks, which were included, after histoprocessing, tissue fragments obtained from biological samples of tumor tissue, parts taken during surgical and microsurgical interventions [5,7].

Stochastic sensors were used as described previously by Stefan-van Staden [4] for the assay of CEA and NSE without any sample (whole blood and tissue) pre-treatment before analysis. Ten minutes were needed to get the diagrams from which first were detected the signatures of CEA and NSE and after that their concentration in whole blood and tissue sample.

## 3. Results and Discussion

Imaging examination was performed (CT scan) for the selected patients—suspects of Microcellular Bronchopulmonary Cancer. Blood was collected (1–5 mL on EDTA coated tubes) and analyzed using the stochastic sensor. Surgery was performed, and the anatomopathological examination of the resection piece (tumor tissue) was done to establish the type of lung cancer, then immunohistochemical analysis of the profile of tumor markers was performed in order to establish the diagnosis. Following these analyses, there was a good correlation between the results obtained using stochastic sensors for NSE in whole blood samples and tumor tests for NSE using the standard immunohistochemical analysis. The most representative images from the pathologic and immunohistochemical examinations performed on the tissue samples analyzed also with the stochastic sensors were shown below. Figure 1. shows central small cell carcinoma of the lung, adjacent to a bronchial gland and hyaline cartilage, H&E stain, 40×.

Figure 2 shows characteristic features of small cell carcinoma of the lung: multiple necrotic areas and Azzopardi effect, H&E stain, 100×.

Figure 3 shows high power view of the neoplastic malignant cells featuring sheets, ribbons, clusters, rosettes or peripheral palisading of small to medium-sized cells with minimal cytoplasm, salt, and pepper chromatin without prominent clumps, hyperchromatic, indistinct nucleoli, nuclear molding, smudging and frequent mitotic figures, H&E stain, 200×.

Figure 4 shows strong cytoplasmic staining for NSE antibody in a case of small cell carcinoma of the lung, IHC stain with DAB chromogen, 200×.

Figure 5 shows small cell lung carcinoma, coloration he, ob. 10×.

Figure 6 shows carcinoma small cell lung “in oatmeal”, Col. He, Lens 40×.

Figure 7 shows NSE, small cell carcinoma, heterogenic positivity, cytoplasmatic, IHC, dab Chromogen, 40× lens.

Within the batch taken in study 43 subjects showed a positive reaction to immunohistochemical analysis for NSE and in two cases, negative reaction.

Whole blood samples and tissue samples were screened using the stochastic sensors. A potential of 125 mV was applied for 20 min and diagrams were recorded. The NSE and CEA were identified in the diagrams accordingly with their signatures [4]; their quantification was also done based on the measurements recorded in the diagrams [4]. The screening method for molecular recognition and quantification of NSE and CEA was proposed earlier by Stefan-van Staden [4].

For the 45 patients considered in this pilot study, all confirmed with microcellular cancers, the following were the results obtained with different methods of analysis: elevated levels of NSE was found in all 45 patients screened with the stochastic sensor, immunohistochemical exam found NSE in 43 tissues, while using ELISA the NSE was found in 40 patients. It was noted that the stochastic sensors were more sensitive, being able to determine very small concentrations of NSE in whole blood and tumor tissue, the level detected being a bit higher than for the other methods because the samples were used as collected from the patients selected by Claudiu Eduard Nistor, and no NSE amount was lost by sample processing; this was an advantage of this technique vs the standard methods (ELISA and IHC) shown also by Stefan-van Staden on her publication [4]. Correlation between the results obtained for NSE using all methods of analysis is shown in Figure 8. As shown in Figure 8, there is a good correlation between data obtained using the three methods. Also, using the stochastic sensors elevated values for CEA were obtained for all patients.

## 4. Conclusions

The comparison of data obtained using the three methods: (1) ELISA kits, (2) the technique developed earlier by Stefan-van Staden—which used stochastic sensors, and (3) IHC shown that the screening method developed earlier by Stefan-van Staden can be validated, being a good alternative for the analysis of biological samples like whole blood and tumor tissue. Furthermore, the screening method provides enough sensitivity to be used in the early diagnosis of microcellular lung cancer. This pilot study had as features utilization of the stochastic sensors and of the screening test for mass screening of population for different lung cancers, and development of a unique screening test for lung cancers; depending on the outcome of the screening test, the patients will be directed for the final diagnostic of the pulmonary tumors to imagistic and anatomopathological and immunohistochemical examination.

Elevated blood levels of NSE and CEA could be a benchmark for detecting, monitoring the therapeutic response and the evolution of patients. The study aimed to develop reliable techniques of rapid diagnosis-screening techniques and early detection of microcellular lung cancer in the risk population, establishing a personalized diagnosis, early initiation of some personalized treatment, increasing the life expectancy of patients with microcellular lung cancer.

## Figures and Tables

**Figure 1 jcm-09-00076-f001:**
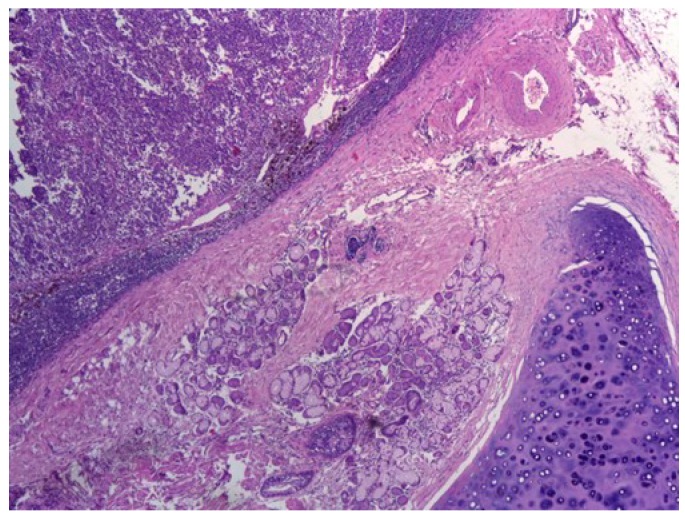
Central small cell carcinoma of the lung.

**Figure 2 jcm-09-00076-f002:**
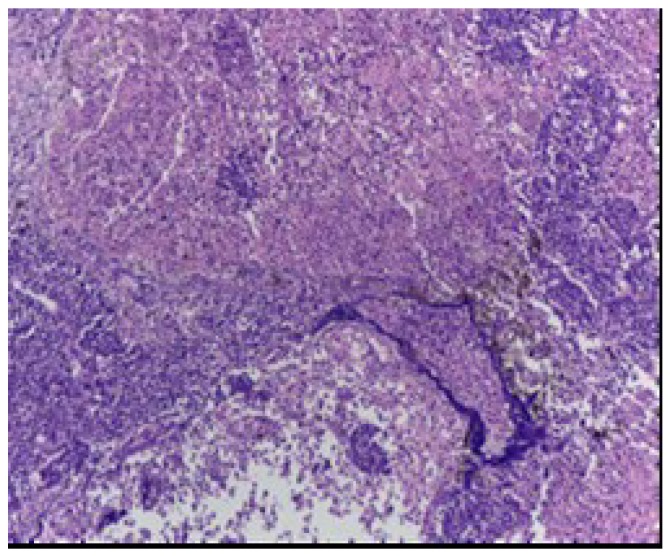
Small cell carcinoma of the lung.

**Figure 3 jcm-09-00076-f003:**
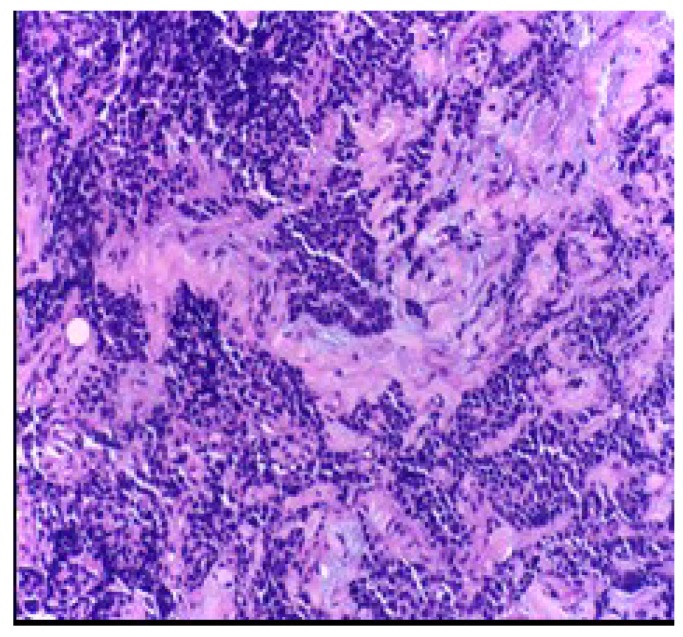
High power view of the neoplastic malignant cells.

**Figure 4 jcm-09-00076-f004:**
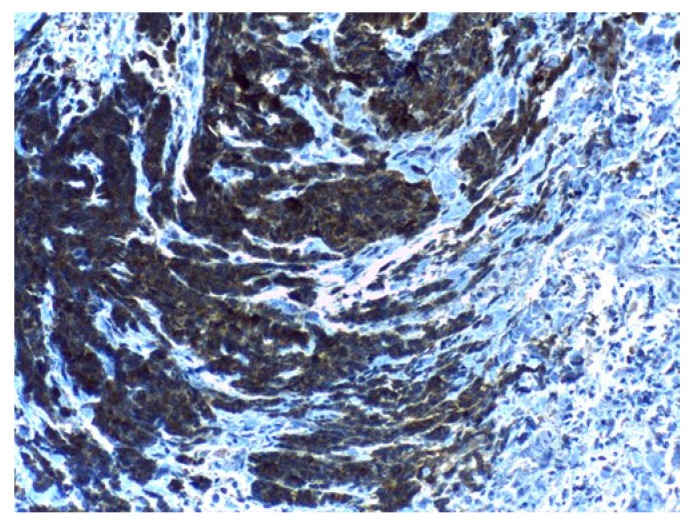
Strong cytoplasmic staining for NSE antibody.

**Figure 5 jcm-09-00076-f005:**
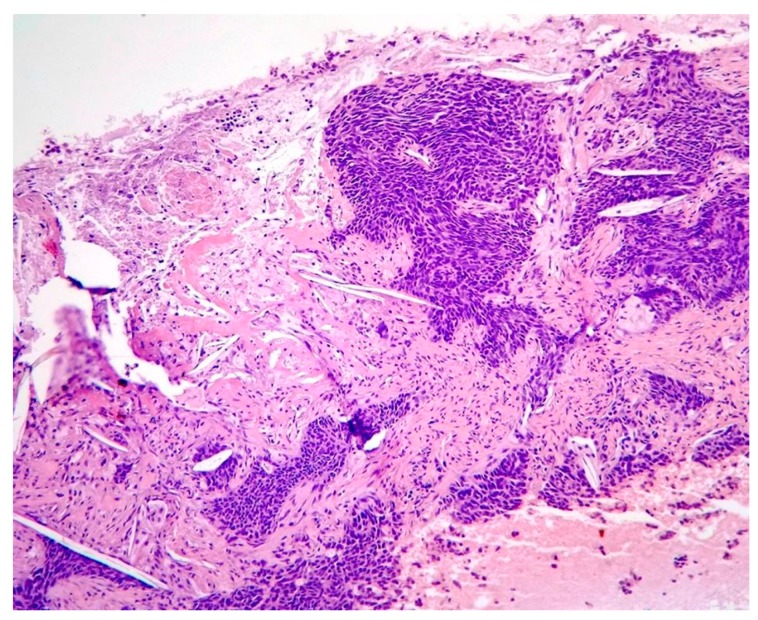
Small cell lung carcinoma.

**Figure 6 jcm-09-00076-f006:**
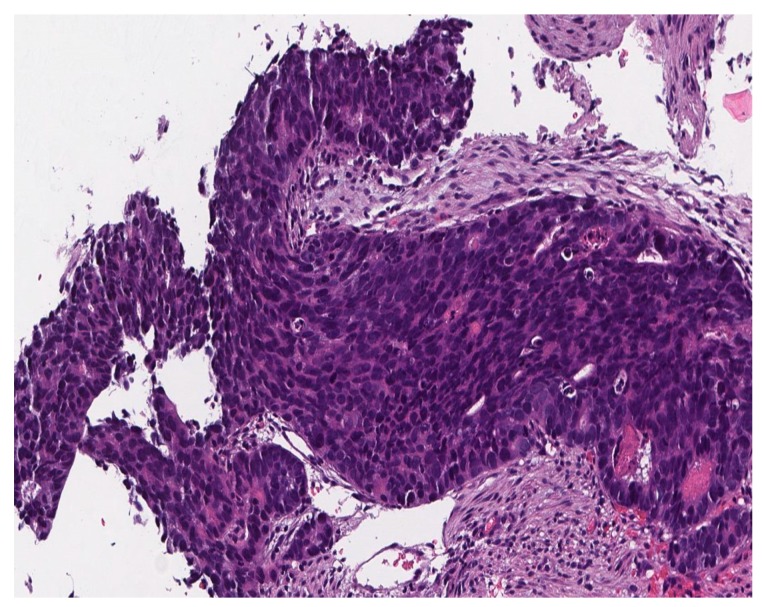
Carcinoma Small cell lung.

**Figure 7 jcm-09-00076-f007:**
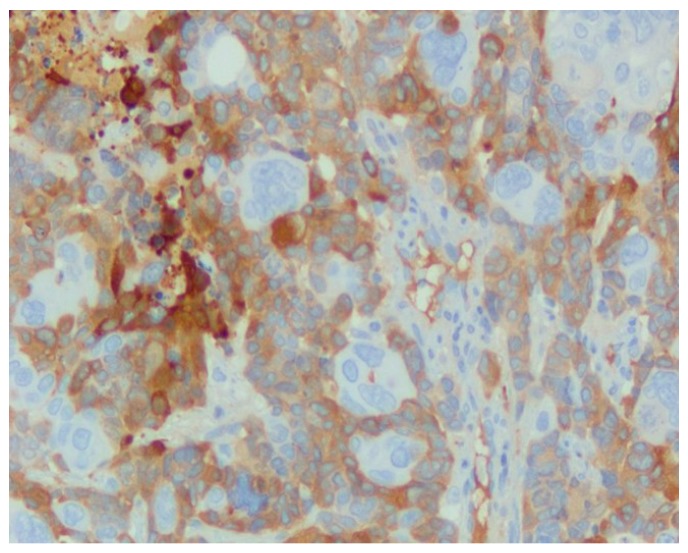
NSE, small cell carcinoma.

**Figure 8 jcm-09-00076-f008:**
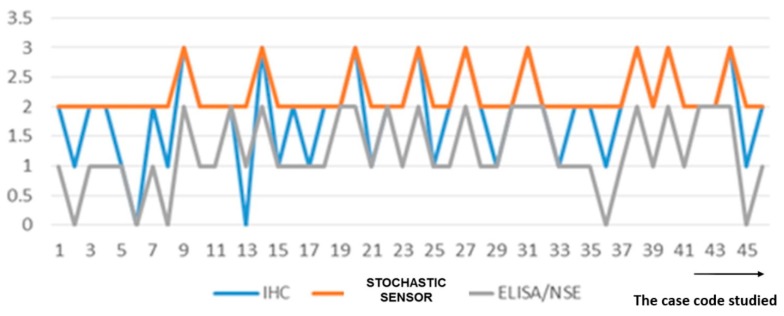
Correlation of the results obtained for the determination of NSE using ELISA, IHC, and the stochastic sensor.

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
