# Peer review of "A Screening Test for Early Diagnosis of Microcellular Bronchopulmonary Cancer-Pilot Study"

_jcm, 2019, doi:10.3390/jcm9010076_

Round 1

Reviewer 1 Report

The manuscript under review is entitled "NSE (Neuron Specific Enolase) Test-Early Diagnosis of Microcellular Bronchopulmonary Cancer by Alternative Method of Analysis-PILOT Study". The technical innovation concerning the development of an alternative method of NSE analysis, i.e. a stochastic biosensor for NSE, and its application to the analysis of whole blood samples from human individuals has already been presented elsewhere (Ref. 3 of the manuscript).  Thus, the main purpose of the present work is to correlate the results obtained after measuring NSE levels in suitable biological samples of 45 patients suspected of  microcellular bronchopulmonary cancer by using the above biosensor, with those obtained by using conventional methodology (ELISA, IHC).

In my opinion, the manuscript has many weak points: e.g. citations concerning diagnosis and specific biomarkers of microcellular bronchopulmonary cancer are rather few; basic information concerning NSE as a tumor marker is not presented as early as in the Introduction part, but rather in the Experimental part (2.6), while literature support is still poor. Another weak point is associated with carcinoembryonic antigen (CEA), which has been introduced on p. 2 as a second (control?) biomarker, making the main purpose of the manuscript somewhat obscure. CEA has been analyzed, along with NSE, with the stochastic biosensor and conventional ELISAs. Despite mentioning both CEA and NSE (p. 7, last paragraph of the Experimental part) as tumor biomarkers for which "a careful analysis of the sensitivity of all the usual detection methods and the correlation of all the obtained results was necessary", the authors do not refer to CEA in the Results and Discussion part -which is, in any case, very short. In general, the manuscript is difficult to follow, due to many typing errors, missing words (there is a short paragraph not written in English, too, p. 7), no numbers for identifying Figures and Tables, etc. To sum up, in my opinion, the manuscript should be written from the scratch, so that the purpose, the results and the conclusions of the research performed can be clearly presented and adequately supported.

Author Response

The manuscript under review is entitled "NSE (Neuron Specific Enolase) Test-Early Diagnosis of Microcellular Bronchopulmonary Cancer by Alternative Method of Analysis-PILOT Study". The technical innovation concerning the development of an alternative method of NSE analysis, i.e. a stochastic biosensor for NSE, and its application to the analysis of whole blood samples from human individuals has already been presented elsewhere (Ref. 3 of the manuscript).  Thus, the main purpose of the present work is to correlate the results obtained after measuring NSE levels in suitable biological samples of 45 patients suspected of  microcellular bronchopulmonary cancer by using the above biosensor, with those obtained by using conventional methodology (ELISA, IHC).

In my opinion, the manuscript has many weak points: e.g.

citations concerning diagnosis and specific biomarkers of microcellular bronchopulmonary cancer are rather few;

We did cited (add) a significant citation in the ref list

basic information concerning NSE as a tumor marker is not presented as early as in the Introduction part, but rather in the Experimental part (2.6), while literature support is still poor.

We did add these information in the Introduction part.

Another weak point is associated with carcinoembryonic antigen (CEA), which has been introduced on p. 2 as a second (control?) biomarker, making the main purpose of the manuscript somewhat obscure. CEA has been analyzed, along with NSE, with the stochastic biosensor and conventional ELISAs. Despite mentioning both CEA and NSE (p. 7, last paragraph of the Experimental part) as tumor biomarkers for which "a careful analysis of the sensitivity of all the usual detection methods and the correlation of all the obtained results was necessary", the authors do not refer to CEA in the Results and Discussion part -which is, in any case, very short. In general, the manuscript is difficult to follow, due to many typing errors, missing words (there is a short paragraph not written in English, too, p. 7), no numbers for identifying Figures and Tables, etc. To sum up, in my opinion, the manuscript should be written from the scratch, so that the purpose, the results and the conclusions of the research performed can be clearly presented and adequately supported.

The Ms was written from scratch – also the title was changed to reflect better the content. CEA got an important role in the diagnosis and treatment of many cancer, and we would like to keep it in the study – not as a significative highly related biomarker for lung cancer, but as a biomarker for the presence of a tumor.

Reviewer 2 Report

This study used “stochastic sensors” to evaluate the association between markers and bronchopulmonary cancer. There’s a lot of novelty in the study concept and design.

Please give some introduction of stochastic sensors so that audience will understand what it is. Please check the grammar in the fourth paragraph of the Introduction. In the first sentence of “Experimental part”, what’s the meaning of “face research”? Please give some introduction of it so that people can understand it more easily. In “Reagents and Materials”, please list the “new instruments” used in this study and their manufacturers. Please cite the figures in the text. Audience can not understand the results if they can not find where the figure is.

Author Response

This study used “stochastic sensors” to evaluate the association between markers and bronchopulmonary cancer. There’s a lot of novelty in the study concept and design.

Please give some introduction of stochastic sensors so that audience will understand what it is. Please check the grammar in the fourth paragraph of the Introduction. In the first sentence of “Experimental part”, what’s the meaning of “face research”? Please give some introduction of it so that people can understand it more easily. In “Reagents and Materials”, please list the “new instruments” used in this study and their manufacturers. Please cite the figures in the text. Audience can not understand the results if they can not find where the figure is.

The MS was re-written for a better understanding, and an intro to stochastic sensors was added.

Round 2

Reviewer 1 Report

The revised version of the manuscript jcm-659978 (new title: "A Screening Test for Early Diagnosis of Microcellular Bronchopulmonary Cancer - Pilot Study") has been significantly improved.

Some minor comments are listed below:

p. 3: 2.6. Determination of CEA and NSE using ELISA, IHC, and stochastic method

A. The text could be shortened, by eliminating general principles of ELISA and IHC techniques. 

B. Could you, please, provide more details about the anti-NSE antibody used in the IHC experiments?

p. 6: Last paragraph, 2nd line

Do you mean "Elevated levels of NSE were found..."?

p. 7: Figure 9

Please, provide suitable headings for X and Y axes.

Clarity will be further improved, if the authors can provide the above mentioned few extra details.

Author Response

The revised version of the manuscript jcm-659978 (new title: "A Screening Test for Early Diagnosis of Microcellular Bronchopulmonary Cancer - Pilot Study") has been significantly improved.

Some minor comments are listed below:

3: 2.6. Determination of CEA and NSE using ELISA, IHC, and stochastic method The text could be shortened, by eliminating general principles of ELISA and IHC techniques.

The paragraph was re-written.

Could you, please, provide more details about the anti-NSE antibody used in the IHC experiments?

-The analysis were performed in the specialised clinical labs accordingly with approved protocols, therefore we do not have access to this data. We only compare the collected data.

6: Last paragraph, 2nd line

Do you mean "Elevated levels of NSE were found..."?

- we did add „elevated levels of ...”

7: Figure 9

Please, provide suitable headings for X and Y axes.

Clarity will be further improved, if the authors can provide the above mentioned few extra details.

-          We did insert the legends for X and Y axes.

Reviewer 2 Report

Figures are still not cited/referred in the text. Only figure 9 is cited/referred in the text. There's no Figure 8??? Audiences can not understand the results if they can not find where the figure is. 

Author Response

-       All figures were cited/ref. in the text. We did correct the number of fig 8.